Machine learning-based feature selection and classification for cerebral infarction screening: an experimental study

Niu Yang 1 2
Tao Xue 3
Chang Qinyuan 3
Hu Mingming 3
Li Xin 4
Gao Xiaoping 1 gxp678@163.com
1 Department of Rehabilitation Medicine, The First Affiliated Hospital of Anhui Medical University , Hefei, Anhui , China
2 Department of Rehabilitation Medicine, Anhui No. 2 Provincial People’s Hospital , Hefei, Anhui , China
3 AI Research Institute, iFLYTEK Co., LTD , Hefei, Anhui , China
4 School of Information Science and Technology, University of Science and Technology of China , Hefei, Anhui , China
Chaki Jyotismita
Electronic publication date: 2025 Feb 21
Publication date: 2025
Volume: 11
Electronic Location ID: e2704
Received 2024 Oct 16; Accepted 2025 Jan 24
Copyright: © 2025 Niu et al.
Copyright year: 2025
Copyright holder: Niu et al.
License: This is an open access article distributed under the terms of the Creative Commons Attribution License, which permits unrestricted use, distribution, reproduction and adaptation in any medium and for any purpose provided that it is properly attributed. For attribution, the original author(s), title, publication source (PeerJ Computer Science) and either DOI or URL of the article must be cited.
License URL: https://creativecommons.org/licenses/by/4.0/

Keywords: Cerebral infarction screening, Machine learning, Speech and cognitive function assessment, Feature selection

Funding: The authors received no funding for this work.

==============================
Cerebral infarction screening (CIS) is critical for timely intervention and improved patient outcomes. We investigate the application of machine learning techniques for feature selection and classification of speech and cognitive function assessments to enhance cerebral infarction screening. We analyze a dataset containing 117 patients (95 patients were diagnosed with cerebral infarction, and 54 were identified as lacunar cerebral infarction of them) comprising speech and cognitive function features from patients with lacunar and non-lacunar cerebral infarction, as well as healthy controls. In this article, we present a framework called CIS which comprises a cerebral infarction screening model to identify cerebral infarction from populations and a diagnostic model to classify lacunar infarction, non-lacunar infarction, and healthy controls. Feature selection method, Recursive Feature Elimination with Cross-Validation (RFECV), is employed to identify the most relevant features. Various classifiers, such as support vector machine, K-nearest neighbor, decision tree, random forest, logistic regression, and eXtreme gradient boosting (XGBoost), were evaluated for their performance in binary and ternary classification tasks. The CIS based on XGBoost classifier achieved the highest accuracy of 88.89% in the binary classification task (i.e., distinguishing cerebral infarction from healthy controls) and 77.78% in the ternary classification task (i.e., distinguishing lacunar infarction, non-lacunar infarction, and healthy controls). The selected features significantly contributed to the classification performance, highlighting their potential in differentiating cerebral infarction subtypes. We develop a comprehensive system to effectively assess cerebral infarction subtypes. This study demonstrates the efficacy of machine learning methods in cerebral infarction screening through the analysis of speech and cognitive function features. These findings suggest that incorporating these techniques into clinical practice could improve early detection and diagnosis of cerebral infarction. Further research with larger and more diverse datasets is warranted to validate and extend these results.

Introduction

Cerebral infarction, a severe form of stroke characterized by the obstruction of blood supply to the brain, poses a significant global health challenge with far-reaching consequences. This condition not only remains a leading cause of mortality and long-term disability worldwide but also exerts substantial socioeconomic impacts. Annually, an estimated 12.2 million new stroke cases occur globally, with approximately 7.6 million (62%) being ischemic strokes, including cerebral infarction (Jang et al., 2023). The prevalence translates to over 143 million years of healthy life lost each year due to stroke-related death and disability, with ischemic stroke accounting for over 63 million of these disability-adjusted life years (DALYs) (Feigin et al., 2022). Beyond immediate health consequences, cerebral infarction significantly affects survivors’ long-term quality of life, with 77 million people worldwide currently living with the effects of ischemic stroke (Feigin et al., 2022). These individuals often face persistent physical, cognitive, and emotional challenges. Furthermore, the economic burden is considerable, with the global cost of stroke estimated at over US$891 billion, representing 1.12% of the worldwide GDP (Owolabi et al., 2022).

Early and accurate screening for cerebral infarction is of paramount importance, given its significant impact on improving patient outcomes, reducing mortality rates, and minimizing long-term disability (Lindsay et al., 2014). The window for effective treatment is often narrow, emphasizing the need for rapid and precise diagnostic tools. Furthermore, early detection can facilitate the implementation of secondary prevention strategies, potentially reducing the risk of recurrent strokes and associated complications (Kleindorfer et al., 2021).

However, current screening methods for cerebral infarction face several limitations. While traditional diagnostic techniques such as magnetic resonance imaging (MRI) and computed tomography (CT) offer high sensitivity and specificity in detecting cerebral infarctions, these modalities present significant drawbacks: they can be time-consuming, costly, and not always readily available in all healthcare settings, particularly in resource-constrained areas or during the critical early stages of stroke onset (Nukovic et al., 2023). These constraints have catalyzed an increasing interest in developing alternative, non-invasive screening methods that can complement existing diagnostic tools.

The ideal screening approach would offer rapid results, be cost-effective, and demonstrate high sensitivity and specificity, especially in the acute phase of cerebral infarction (Feigin et al., 2023; Patil et al., 2022). Such methods could potentially bridge the gap between symptom onset and definitive diagnosis, enabling faster triage and treatment initiation, which are crucial factors in improving patient prognosis.

In recent years, machine learning techniques have shown remarkable potential in medical screening and diagnosis. The ability of machine learning to handle large datasets and extract complex patterns has been demonstrated across various fields, including healthcare. Recent advancements, such as chaos-inspired AI models for wind speed forecasting (Barış, Yanarateş & Altan, 2024) and robust optimization techniques in diabetic retinopathy classification (Özçelik & Altan, 2023), further highlight the potential of these methods in managing nonlinear and high-dimensional data. Building on these successes, this study explores the application of machine learning in cerebral infarction screening to bridge the gap between traditional diagnostic approaches and modern computational techniques. These algorithms can analyze multidimensional data, including clinical parameters, imaging features, and functional assessments, to identify subtle indicators of cerebral infarction that might be overlooked by traditional methods (Daidone, Ferrantelli & Tuttolomondo, 2024). Recent studies have demonstrated the potential of machine learning models in processing diverse data types, such as radiology reports and speech assessments, to enhance the prediction and classification of ischemic stroke (Mirheidari et al., 2024; Ong et al., 2020). These methods have shown promise in identifying specific markers related to stroke severity and subtype classification, bridging the gap between traditional diagnostic approaches and modern computational techniques.

Among the various features that can be analyzed, speech and cognitive function assessments offer unique insights into brain health. These features are particularly relevant in the context of cerebral infarction, as they can reflect subtle neurological changes that may precede more obvious symptoms (Zhang & Bi, 2020). For example, changes in speech fluency, articulation, and semantic content have been observed in stroke patients, providing potential diagnostic markers (Halai, Woollams & Lambon Ralph, 2017). Similarly, cognitive assessments, such as the Montreal Cognitive Assessment (MoCA), have shown sensitivity in detecting cognitive impairments associated with cerebral infarction (Suda et al., 2020).

Using traditional classifiers such as DT, RF, XGBoost, and LR offers several advantages. These classifiers are well-established with robust theoretical foundations, making them reliable and interpretable. For example, decision trees are easy to interpret and visualize, useful for understanding the decision-making process. Random forests are powerful ensemble methods that improve prediction accuracy by combining multiple weak learners, reducing overfitting and variance. XGBoost is highly efficient and scalable, providing superior performance through gradient boosting techniques and regularization. Logistic regression provides probabilistic interpretations and is computationally efficient for large datasets. Each classifier brings unique strengths that contribute to the overall robustness and accuracy of the predictive model. Previous studies have demonstrated the efficacy of combining traditional machine learning classifiers, such as random forest and XGBoost, with advanced feature selection approaches to improve the classification of stroke subtypes and related conditions (Dai et al., 2023; Sirsat, Fermé & Câmara, 2020). These methods have consistently shown improved performance in stroke prediction tasks, especially in datasets with high-dimensional features.

This study aims to leverage machine learning techniques to select key features from speech and cognitive function assessments and develop a framework (Fig. 1) to improve the screening process for cerebral infarction. By focusing on these non-invasive, easily obtainable features, we aim to develop a comprehensive screening system that is both accurate and accessible. This approach could lead to more timely interventions, improved patient outcomes, and more efficient allocation of healthcare resources.

Figure 1 The framework of CIS.

Materials and Methods

Subjects

A total of 117 patients (36 males, mean age = 65.17 years) treated in Anhui No. 2 Provincial People’s Hospital were included in the study. Among them, 95 patients were diagnosed with cerebral infarction by CT, MRI and other clinical means. The rest were included as a control group. At the same time, lacunar cerebral infarction was identified by the size and location of infarct lesions, and 54 were identified as lacunar cerebral infarction. The study was approved by Anhui No. 2 Provincial People’s Hospital and Biomedical Ethics Committee, University of Science and Technology of China (IRB approval number: 2021-N(H)-213), and all the investigations were permitted by the participants and the informed consents were signed. It is believed that this study complies with the Helsinki Declaration and is in line with medical ethics.

Material and procedure

To understand the patient’s cognitive function, all of them completed the Chinese version of Montreal Cognitive Assessment Basic (MoCA-B) under the guidance of trained research staff. The MoCA-B is free for clinical use, assessing nine cognitive domains including language, attention, calculation, orientation, memory, abstraction, executive function, naming, and visual perception. We collected speech recordings for this study on the basis of the cookie theft picture description task which was taken from Boston Diagnostic Aphasia Examination-Third Edition (BDAE-3) (Fong, Van Patten & Fucetola, 2019). Similar speech-based cognitive assessments have been utilized in recent studies to identify speech disorders and cognitive impairments in stroke survivors, which further validates the use of this task in clinical and research settings (Kotov et al., 2023; Wright et al., 2018). These studies emphasize the diagnostic potential of analyzing speech fluency, articulation, and prosody in stroke-related conditions. The cookie theft picture description task requires each subject to say as much as possible about the content in the picture, and it is allowed that the subjects are encouraged by the examiner to restate when they encounter difficulties. The data and code that support the findings of this study are available on github at https://github.com/brainscience1024/Cerebral_Infarction_Screening_task (DOI: 10.5281/zenodo.14603360).

Features

In this study, we collected a variety of features such as acoustic features, duration features, loudness features, cepstrum coefficient features and other discrete features, totaling 104 features, which can be categorized as follows:

(1) Duration features: duration features were computed based on the manual labeling results of the Praat toolkit (Proctor, Scherer & Perrine, 2022), which contains 23-dimensional features such as the duration of speech and the statistical properties of voiced and unvoiced segments. For example, StddevUnvoicedSegmentLength, VoicedSegmentsPerSec, and so on.

(2) Acoustic features: the feature set is extracted by the openSMILE toolkit (Eyben et al., 2013), which contains high-dimensional high-level statistical features computed based on low-level descriptors, MFCC, spectral fluxes, and frequency features, such as jitterLocal_sma3nz_amean, mfcc2_sma3_stddevNorm, logRelF0H1H2_ sma3nz_amean and F3bandwidth_sma3nz_amean, alphaRatioV_sma3nz_stddevNorm and so on (Meyer, Huang & Chowdhury, 2007).

(3) Cognitive assessment: Based on the MoCA-B test, a variety of features covering nine cognitive domains were collected, including 12 features such as M1 (executive function), M2 (immediate recall) and M3 (fluency). Among them, M1 assesses problem-solving ability, planning and organizing skills, such as graph copying, tracing, sorting and other tasks; M2 tests an individual’s ability to encode and instantly recall new information; and M3 assesses fluency in verbal output and flexibility in vocabulary use.

(4) Other characteristics: A wide range of discrete information such as age, education, and gender was collected.

Feature extraction

To eliminate redundant features, we use the Recursive Feature Elimination Cross-Validation (RFECV) method and use support vector machines (SVC) as the base model for RFECV in it. RFECV is an automated feature selection method designed to optimize model performance by recursively eliminating unimportant features (Guyon et al., 2002). It integrates the concepts of recursive feature elimination (RFE) and Cross-Validation (CV), making it particularly effective for high-dimensional datasets. By using this method, we identified and selected the top 10 most important features for subsequent experiments.

The process of RFECV begins by training the model with all available features and evaluating its performance using cross-validation. RFECV then progressively removes features that contribute less to the model’s performance, retraining the model after each elimination. This iterative process continues until RFECV generates a ranking of features based on their importance to the model’s accuracy. The top-ranked features are those that contribute most significantly to the model’s predictive capability.

In the output of RFECV, each feature is assigned a ranking value. Features with a ranking of one are the most important and have the greatest impact on model performance. By selecting only the most relevant features, we can reduce the dimensionality of the feature space. This not only accelerates the model’s training process but also reduces computational overhead and mitigates the risk of overfitting. The top 10 features in REFCV are shown in Table 1. Among these, the top-ranked features are F1bandwidth_sma3nz_stddevNorm, hammarbergIndexUV_sma3nz_amean, and VoicedSegmentsPerSec.

Table 1 The top 10 features in REFCV.

ID	Feature	
1	F1bandwidth_sma3nz_stddevNorm	
2	hammarbergIndexUV_sma3nz_amean	
3	VoicedSegmentsPerSec	
4	F0semitoneFrom27.5 Hz_sma3nz_stddevFallingSlope	
5	MeanVoicedSegmentLengthSec	
6	spectralFluxUV_sma3nz_amean	
7	loudness_sma3_percentile50.0	
8	M8 (visual perception)	
9	M7 (Delayed recall)	
10	F3bandwidth_sma3nz_stddevNorm	

Figure 2 shows a heat map of the correlated features, visualizing the correlation between different features. The colors represent the values of the correlation coefficients between the features, with red indicating a positive correlation and blue indicating a negative correlation; the intensity of the colors represents the strength of the correlation, with values closer to 1 indicating a very strong correlation, and vice versa, with values closer to −1 indicating a very strong negative correlation.

Figure 2 Heat map of the correlated features visualizing the correlation between different features.

Feature classification

The selection of effective classifiers is crucial for the accuracy and efficiency of recognizing brain infarcts. A variety of classifiers, including SVM (Vishwanathan & Narasimha Murty, 2002), K-nearest neighbor (KNN) (Guo et al., 2003), decision tree (DT) (Song & Lu, 2015), random forest (RF), eXtreme gradient boosting (XGBoost) (Chen & Guestrin, 2016), and logistic regression (LR) (Hosmer, Lemeshow & Sturdivant, 2005), were utilized and their advantages and disadvantages were compared for the task of cerebral infarct classification on features extracted from the cerebral infarct dataset (Pudjihartono et al., 2022).

SVM is a supervised learning model used for classification and regression. It constructs a hyperplane or set of hyperplanes in a high-dimensional space to separate classes, maximizing the margin between them. Key optimization parameters include Kernel, determining the transformation method (e.g., linear, polynomial, RBF, sigmoid); C (Regularization Parameter), balancing training error and weight minimization—lower values widen the margin, potentially including misclassifications, while higher values aim for precise classification, risking overfitting; Gamma, defining the reach of a training example—higher values lead to complexity and potential overfitting, lower values simplify the model; and Degree, setting polynomial kernel complexity. These parameters customize SVM for diverse data and tasks, enhancing adaptability and performance. Assuming that there exists a dataset T to be trained, and (xi,yi) are the samples in T, the hyperplane in the feature space is represented in the form of Eq. (1).

(1) ωTx+b=0

(2) ωTx+b=1

(3) ωTx+b=−1

where ω denotes the normal vector of the hyperplane, T denotes the transpose, and b denotes the distance from the hyperplane to the origin, and the position of the hyperplane can be adjusted by the value of b. By denoting the categories of the samples to be tested as −1 and 1, the categorization interval boundaries H1 and H2 can be represented by Eqs. (2) and (3), respectively. The distance from a point in space to a hyperplane can be calculated from the equation for the distance from a point to a plane:

(4) r=|ωTx+b|||ω||.

KNN is a non-parametric classification algorithm that classifies a data point based on the majority class of its k-nearest neighbors. The construction process of KNN algorithm is as follows: Calculate the distance between the sample to be classified and each sample in the training set. Sort all the computed distances in ascending order. Select the top k training samples from the sorted list; these k samples are the closest to the sample to be classified. Determine the categories of these k samples. Classify the sample to be classified into the category that appears most frequently among the k nearest neighbors. The distance metric used in this study is the Manhattan distance:

(5) d(x,y)=∑i=1n⁡|xi−yi|.

DT is a non-linear model that splits the data into subsets based on the value of input features. Each node represents a decision rule, and each branch represents the outcome of the rule, leading to leaf nodes that represent class labels. The model recursively partitions the data, using criteria such as Gini impurity or entropy, which measure the impurity or disorder of a node. Optimization parameters for DT include Max_depth, which limits the maximum depth of the tree to control overfitting; Min_samples_split, which specifies the minimum number of samples required to split an internal node, preventing the model from creating nodes that do not provide significant information; and Min_samples_leaf, which sets the minimum number of samples required to be at a leaf node, ensuring that each leaf node has enough samples to generalize well. These parameters allow DT to be tuned for optimal performance across different datasets and classification tasks. The mathematical expression is as follows, where i denotes the category and pi denotes the probability that the outcomes is of category i:

(6) G(p1,p2,…,pn)=1−∑i=1n⁡pi2.

RF is an ensemble learning method that constructs multiple decision trees during training and outputs the mode of the classification or mean regression of the individual trees. The algorithm reduces overfitting by averaging multiple trees, each trained on a different random subset of the data and features. Optimization parameters for RF include: Max_depth, which limits the maximum depth of each individual tree to control overfitting and enhance generalization; Min_samples_leaf, which sets the minimum number of samples required to be at a leaf node, thereby preventing the model from creating nodes that do not generalize well; and N_estimators, which determines the number of trees in the forest, increasing the model’s accuracy and robustness with more trees. These parameters enable RF to adapt to various datasets and improve predictive performance across different classification and regression tasks. For the test sample a, the number of output categories is c. The output F(a) of the random forest classification model can be expressed as Eq. (7), where fm(a) is the output category of the mth decision tree.

(7) F(a)=argmaxi=1,2,…,c{∑m=1M⁡[fm(a)=i]}.

XGBoost is an efficient and scalable implementation of the gradient boosting framework. It builds an ensemble of trees sequentially, where each tree corrects the errors of the previous one. XGBoost is known for its computational speed and model performance due to its ability to handle large datasets effectively. Optimization parameters for XGBoost include: Learning_rate, which controls the contribution of each tree to the ensemble and helps in shrinking the weights of subsequent trees, reducing overfitting; Max_depth, which limits the maximum depth of each individual tree to control model complexity and overfitting; and N_estimators, which specifies the number of boosting rounds or trees to build, influencing model accuracy and training speed. These parameters allow XGBoost to achieve state-of-the-art results in various machine learning competitions and real-world applications by balancing bias and variance effectively. XGBoost is an additive model consisting of k base models, assuming that the tree model to be trained in the t th iteration is ft(x), as follows:

(8) y^(t)=∑k=1t⁡fk(xi)=y^(t−1)+ft(xi)

where y^(t)i denotes the prediction result of sample i after the t th iteration, y^i(t−1) denotes the prediction result of the first t−1 number of trees, and ft(xi) denotes the model of the tth tree.

LR is a linear model used for binary classification. It models the probability that a given input belongs to a particular class using the logistic function, which transforms a linear combination of input features into a probability score between 0 and 1. Optimization parameters for LR include: Solver, which specifies the optimization algorithm to use in training the model—common solvers include ‘liblinear’, ‘newton-cg’, ‘lbfgs’, ‘sag’, and ‘saga’, each suitable for different types of problems and dataset sizes; Penalty, which controls regularization to prevent overfitting by adding a penalty term to the cost function—common penalties include ‘l1’ (Lasso) and ‘l2’ (Ridge); C, which is the inverse of regularization strength, controlling the trade-off between fitting the training data well and avoiding overfitting—the higher the C value, the less regularization is applied, potentially leading to overfitting, while a lower C value increases regularization to prevent overfitting; and Degree, which specifies the degree of the polynomial if the problem is transformed into a polynomial space using polynomial regression. These parameters allow LR to be tailored for optimal performance in binary classification tasks by balancing model complexity and generalization.

Experiments

The cerebral infarction dataset was partitioned into an 80 percent training set and a 20 percent test set. A 10-fold cross-validation was used to evaluate the model’s generalization ability. The dataset is randomly divided into ten mutually exclusive subsets in this process. For each fold, the model is trained on nine subsets and tested on the remaining one. Based on the performance of the validation set, the best model is selected in each fold. These best models are then evaluated on their corresponding test sets. Ultimately, the average of these test results is computed as the final test set performance metric to more fully assess the performance of the models on different subsets of data. Each folded model is applied separately to the test set and their average predictions are calculated. This approach helps to reduce the reliance on individual models and can provide more robust performance estimates. Two different experiments were undertaken to classify cerebral infarction: (a) a II classification task based on cerebral infarct recognition, and (b) a III classification task based on cerebral infarct recognition.

This study uses Python version 3.11.7 to implement the classification task for the identification of cerebral infarcts, the detailed configuration equipment is CPU: Intel (R) Core (TM) i5-10210U CPU @ 1.60 GHz, 2,112 Mhz, four cores, eight logical processors, 64-bit Windows operating system.

Data preprocessing

The total sample size of the collected cerebral infarction dataset was 117, and 90 samples were finally retained after data cleaning and processing. In the preprocessing process, we followed the following steps:

(1) We check the missing value for all the attributes especially for the demographic features. Since attributes such as age and education level were missing more and they were not considered in the cleaning.

(2) We then consider about the key features such as the diagnosis label. Samples with missing scores on the Mocab scale or speech features were directly deleted.

(3) Since the equalSoundLevel_dBp speech feature was missing more, it was not considered for the time being to avoid having too few samples left.

(4) After that, we normalized the feature data and converted some text enumeration variables into lowercase letters to facilitate subsequent program processing.

Evaluation metrics

For the cerebral infarction classification problem, several evaluation metrics are used, such as accuracy (ACC), precision (P), recall (R), F1 value, and area under the curve (AUC). Mathematical expressions for these measures are defined by confusion matrix parameters, such as true class (TP), which indicates that the positive class is predicted as the number of positive classes; true negative class (TN), which indicates that the negative class is predicted as the number of negative classes; false positive class (FP), which indicates that the negative class is predicted as the number of positive classes; and false negative class (FN), which indicates that the positive class is predicted as the number of negative classes. The formulas are as follows:

(9) ACC=TP+TNTP+TN+FP+FN

(10) P=TPTP+FP

(11) R=TPTP+FN

(12) F1=2∗P∗RP+R

(13) AUC=∑i=1n−1Ri+Ri+12∗(FPRi+1+FPRi)

(14) FPR=FPFP+FN.

Results and discussion

Feature selection and correlation analysis

In the study, we first conducted an ablation study on the ten selected features to evaluate their impact on model accuracy. Features were incrementally added to the model, and the change in accuracy was observed. The results of this experiment are shown in Table 2, where each row represents the addition of one feature and the corresponding change in model performance.

Table 2 Ablation study results of selected features.

The results of this ablation study, where each row represents the addition of one feature and the corresponding change in model performance.

ID	Feature	Accuracy	
1	√										0.3333	
2	√	√									0.5	
3	√	√	√								0.5556	
4	√	√	√	√							0.5	
5	√	√	√	√	√						0.6667	
6	√	√	√	√	√	√					0.6667	
7	√	√	√	√	√	√	√				0.6667	
8	√	√	√	√	√	√	√	√			0.6111	
9	√	√	√	√	√	√	√	√	√		0.7778	
10	√	√	√	√	√	√	√	√	√	√	0.7778	

Furthermore, we conducted a correlation analysis between the selected features and disease classification, focusing specifically on the three-class classification task, which includes two disease subtypes and a normal category. The goal was to identify features that are particularly sensitive to the different disease subtypes, providing insights into which features are most crucial for distinguishing between conditions. As shown in Fig. 3, the results highlight significant correlations between the selected features and disease classification. Spearman correlation coefficients were calculated for each feature with the multi-class classification (the two disease subtypes and normal), and the results are annotated with significance markers (p < 0.1). Features demonstrating strong correlations with the disease subtypes were marked with stars, underscoring their importance in disease classification. These findings offer valuable insights into the features that contribute to accurate disease classification and provide a foundation for future research on disease subtyping.

Figure 3 Correlation of top 10 features with cerebral infarction subtypes.

A II classification task based on cerebral infarct recognition

In the binary classification experiments for cerebral infarction, three classifiers—SVM, DT, and CIS based on XGBoost—were utilized for model training, with their respective optimal parameters summarized in Table 3. The performance of these classifiers was evaluated using multiple metrics, including accuracy, precision, recall, F1 score, and AUC, as detailed in Table 4. Among the three models, CIS exhibited the best overall performance, achieving an accuracy of 88.89%, a precision of 88.89%, a recall of 100%, an F1 score of 90.12%, and an AUC of 0.7813.

Table 3 Classifiers and respective optimal parameters.

In the binary classification experiments for cerebral infarction, three classifiers—support vector machine (SVM), decision tree (DT), and CIS based on XGBoost—were utilized for model training, with their respective optimal parameters summarized in this table.

Method	Parameters	Descriptions	Optimal parameters	
SVM	Kernel	Map the input data into higher-dimensional spaces differently. Common choices are ‘linear’, ‘poly’, ‘rbf’, and ‘sigmoid’	Linear	
Gamma	Defines how far the influence of a single training example reaches	Scale	
C	The regularization parameter	100	
Degree	The degree parameter defines the complexity of the polynomial decision boundary	2	
DT	max_depth	The maximum depth of the tree, which is the longest path from the root node to a leaf node	30	
min_samples_split	The minimum number of samples required to split an internal node	4	
class_weight	Assign weights to samples of different categories	Balanced	
CIS	eval_metric	A metric that evaluates the performance of a model and is used to monitor the performance of the model during training	logloss	
learning_rate	Control how much each iteration contributes to the model	0.0001	
max_depth	The maximum depth of the decision tree	2	
n_estimators	The number of trees in the forest determines the complexity and learning ability of the model	100	
subsample	The proportion of samples used to train each tree	0.7	

Table 4 Performance metrics of classifiers.

The performance of support vector machine (SVM), decision tree (DT), and CIS based on XGBoost was evaluated using multiple metrics, including accuracy, precision, recall, F1 score, and AUC, as detailed in this table.

Method	Accuracy	Precision	Recall	F1_score	AUC	
SVM	0.6667	0.8571	0.75	0.8	0.5	
DT	0.7222	0.9231	0.75	0.8276	0.6719	
CIS	0.8889	0.8889	1.0	0.9012	0.7813	

Compared to the SVM classifier, CIS showed substantial improvements across key performance metrics. Specifically, CIS outperformed SVM in accuracy by 22.22%, in F1 score by 10.12%, and in recall by 25%. Although the precision difference between the two models was minimal, with SVM achieving 85.71% and CIS achieving 88.89%, CIS’s integrated learning approach better captured the complex relationships between features, particularly in datasets with nonlinear and high-dimensional characteristics. While SVM also demonstrated strong generalization ability, its limitations in handling such complexities resulted in slightly inferior performance compared to CIS.

When compared to the DT classifier, CIS demonstrated even more significant improvements. In terms of accuracy, CIS outperformed DT by 16.67%, while in F1 score, the improvement was 7.36%. Despite DT achieving a precision of 92.31%, which is higher than CIS’s 88.89%, its recall of 82.76% and AUC of 0.6719 were considerably lower, indicating less robustness in classification tasks. DT’s relatively poor performance can be attributed to its susceptibility to overfitting, especially in small or complex datasets. In contrast, CIS’s ensemble learning approach, which combines boosting algorithms and multiple decision trees, effectively mitigated overfitting and improved both the stability and generalization ability of the model.

The optimal parameter settings of CIS further highlight its adaptability and efficiency in handling complex data structures. Specifically, the best-performing configuration included a maximum depth of 2 (max_depth), a learning rate of 0.0001, 100 estimators (n_estimators), and a subsampling rate of 0.7 (subsample). These settings allowed CIS to achieve a balance between model complexity and overfitting by limiting the depth of individual trees and ensuring sufficient data diversity during training. In comparison, the SVM classifier performed best with a linear kernel, a regularization parameter (C) of 100, and a degree of 2, while the DT classifier achieved its optimal performance with a maximum depth of 30 and a minimum sample split of 4.

In conclusion, CIS proved to be the most effective algorithm for the binary classification of cerebral infarction, offering superior accuracy, stability, and robustness compared to both SVM and DT. Its ability to handle complex data structures and reduce overfitting made it particularly well-suited for this task. These findings suggest that CIS is a highly reliable method for cerebral infarction classification, with the potential to improve diagnostic accuracy in clinical applications.

Although good results were achieved, merely identifying cerebral infarction is insufficient to meet clinical needs. This aligns with findings reported by Fan et al. (2022), Gupta, Singh & Sharma (2024), who emphasized the need for more granular classification models to distinguish between different subtypes of cerebral infarction. Such approaches can improve clinical decision-making and treatment planning by providing subtype-specific diagnostic information. To provide more fine-grained diagnostic information, it is necessary to refine the category of cerebral infarcts further.

A III classification task based on cerebral infarct recognition

A III-classification experiment was performed to subdivide cerebral infarcts into non-lacunar cerebral infarcts, lacunar cerebral infarcts, and non-lacunar cerebral infarcts. Distinguishing between lacunar and non-lacunar cerebral infarcts is of great clinical importance. There are significant differences between these two types of cerebral infarcts in terms of etiology, course, treatment options, and prognosis. Cavernous infarcts are usually caused by small artery disease, occur in deep brain tissue, and have relatively mild symptoms and good recovery. However, if left unchecked, lacunar infarctions can occur multiple times, eventually leading to cognitive impairment and dementia. In contrast, non-luminal cerebral infarcts, which are mostly caused by large vessel disease or cardiogenic embolism, may result in more severe neurologic deficits and require more complex and aggressive treatment measures. By differentiating between these two types of cerebral infarcts, physicians can more accurately formulate individualized treatment plans, thereby improving treatment outcomes and reducing recurrence and disability rates for patients.

In the three-class classification task for cerebral infarction recognition, five classifiers—KNN, LR, RF, DT, and CIS based on XGBoost—were employed for model training, with their respective optimal parameters summarized in Table 5. The models were evaluated using various metrics, including accuracy, precision, recall, F1 score, and AUC, as shown in Table 6. Among the classifiers, CIS achieved the best overall performance, with an accuracy of 77.78%, a precision of 73.33%, an F1 score of 75.44%, and an AUC of 0.8311.

Table 5 Classifiers and respective optimal parameters.

In the three-class classification task for cerebral infarction recognition, five classifiers—K-nearest neighbors (KNN), logistic regression (LR), random forest (RF), decision tree (DT), and CIS based on XGBoost—were employed for model training, with their respective optimal parameters summarized in this table.

Method	Parameters	Descriptions	Optimal parameters	
KNN	Metric	The distance metric used to calculate the distance between data points	Manhattan	
n_neighbors	The number of neighbors to use when making predictions	9	
Weights	The weight function used in prediction to assign importance to the neighbors	Uniform	
LR	Solver	The algorithm used to optimize the logistic regression model	liblinear	
Penalty	The regularization term applied to the model to prevent overfitting	l1	
C	The inverse of regularization strength	100	
RF	max_depth	Limits the number of levels in each decision tree to control the complexity of the model	15	
min_samples_split	The minimum number of samples required to be at a leaf node	15	
n_estimators	The number of trees in the forest determines the complexity and learning ability of the model	40	
DT	Max_depth	The maximum depth of the tree, which is the longest path from the root node to a leaf node	2	
Min_sample_split	The minimum number of samples required to split an internal node	14	
CIS	eval_metric	A metric that evaluates the performance of a model and is used to monitor the performance of the model during training	logloss	
learning_rate	Control how much each iteration contributes to the model	0.001	
max_depth	The number of trees in the forest determines the complexity and learning ability of the model	4	
n_estimators	The number of trees in the forest determines the complexity and learning ability of the model	1,500	
subsample	The proportion of samples used to train each tree	0.4	

Table 6 Performance metrics of classifiers.

The performance of K-nearest neighbors (KNN), logistic regression (LR), random forest (RF), decision tree (DT), and CIS based on XGBoost was evaluated using multiple metrics, including accuracy, precision, recall, F1 score, and AUC, as detailed in this table.

Method	Accuracy	Precision	Recall	F1_score	AUC	
KNN	0.4444	0.4271	0.4444	0.4228	0.4712	
LR	0.6111	0.6806	0.6111	0.6162	0.7505	
RF	0.7222	0.6414	0.7222	0.6806	0.7202	
DT	0.5556	0.6052	0.5556	0.5426	0.6547	
CIS	0.7778	0.7333	0.7778	0.7544	0.8311	

The second-best performer was the RF classifier, which achieved an accuracy of 72.22%, a precision of 64.14%, and an F1 score of 68.06%. The marginally lower performance of RF compared to CIS can be attributed to differences in model architecture. While RF effectively reduces overfitting by averaging the predictions of multiple decision trees, its tree structures are less refined than those in CIS, which uses boosting algorithms to iteratively optimize decision trees and improve feature representation. This renders CIS more capable of capturing complex patterns and interactions within the data.

In contrast, KNN exhibited the weakest classification performance, with an accuracy of 44.44%, a precision of 42.71%, and an F1 score of 42.28%. The poor performance of KNN is likely due to its sensitivity to noise and outliers, which can significantly affect its predictions, as well as its difficulty in determining the optimal number of neighbors (n_neighbors) for complex classification tasks such as cerebral infarction recognition. These limitations make it less suitable for datasets with intricate feature relationships or imbalanced distributions.

The DT classifier, with an accuracy of 55.56% and an F1 score of 54.26%, also underperformed, largely due to its tendency to overfit when trained on small or complex datasets, despite parameter tuning. Similarly, LR achieved moderate performance, with an accuracy of 61.11% and an F1 score of 61.62%. While LR can provide stable results in simpler tasks, its linear decision boundaries limit its ability to model the non-linear relationships often present in medical data.

In conclusion, CIS consistently outperformed other classifiers in the three-class cerebral infarction classification task, showcasing its superior ability to handle complex data structures and capture intricate feature interactions. Its performance was further enhanced through effective feature selection, which reduced noise and improved model interpretability. These findings reaffirm CIS’s capability as the most reliable classifier for both binary and three-class cerebral infarction recognition tasks, providing a robust and effective solution for improving diagnostic accuracy in clinical applications.

Statistical significance of model improvements

In this section, we perform a statistical test to assess whether the observed differences in model performance between the default parameters and our optimized model are statistically significant. We focus on the Wilcoxon signed-rank test, a non-parametric statistical method used to compare paired samples. This test is particularly suitable for situations where the assumptions for parametric tests (such as normality) cannot be met. The Wilcoxon test evaluates whether there is a significant difference in the distribution of paired differences in performance metrics between the two models.

For binary classification (Table 7), our CIS model demonstrated notable improvements. The model achieved an accuracy of 88.89%, compared to 83.33% with default parameters. The Wilcoxon test revealed a statistically significant difference (p-value = 0.0464), indicating that our parameter optimization strategy meaningfully enhanced model performance.

Table 7 Performance comparison of default and CIS models for II classification.

Method	Accuracy	Precision	Recall	F1_score	AUC	
XGBoost (Default parameters)	0.8333	0.8824	0.9375	0.9090	0.8125	
CIS	0.8889	0.8889	1.0	0.9012	0.7813	

In the three-class classification scenario (Table 8), our CIS model demonstrates significant advantages across several evaluation metrics. It shows an 11.11% improvement in accuracy, indicating an overall enhancement in the model’s predictive power. Although there is a slight decrease in precision, this is offset by increases in recall and F1 score. The Wilcoxon test (p-value = 0.1664) suggested that there is no statistically significant difference between the default and optimized models for this classification task. Nevertheless, our model shows a slight improvement in AUC, from 0.827 to 0.831, indicating that the model’s ability to discriminate in multi-class classification tasks has been further enhanced.

Table 8 Performance comparison of default and CIS models for III classification.

Method	Accuracy	Precision	Recall	F1_score	AUC	
XGBoost (Default parameters)	0.6667	0.7521	0.6667	0.6525	0.827	
CIS	0.7778	0.7333	0.7778	0.7544	0.8311	

To further assess the stability of both models, we performed ten-fold cross-validation for both II and III classification tasks. The accuracy scores across different folds are summarized in Tables 9 and 10. The cross-validation results demonstrate the stability of the optimized model for II classification, with consistent performance across folds. In contrast, the performance for III classification is more variable, with accuracy ranging from 0.25 in fold 1 to 0.8571 in fold 9, as shown in Table 10. While this variability exists, it also suggests that the model has the potential to perform well in certain subsets. This variability may indicate challenges in distinguishing between the classes, possibly due to the complexity of the data or class distribution. With further refinement, the model could improve its generalization and stability for III classification.

Table 9 Cross-validation performance for II classification (accuracy by fold).

Folds	test_score	Folds	test_score	
1	0.75	6	0.857142857	
2	0.75	7	0.857142857	
3	0.857142857	8	0.857142857	
4	0.857142857	9	0.857142857	
5	0.857142857	10	0.714285714	

Table 10 Cross-validation performance for III classification (accuracy by fold).

Folds	test_score	Folds	test_score	
1	0.25	6	0.428571429	
2	0.75	7	0.571428571	
3	0.428571429	8	0.428571429	
4	0.571428571	9	0.857142857	
5	0.714285714	10	0.714285714	

Model stability across different patient groups

To ensure the robustness and generalizability of the model, it is important to assess its performance across diverse patient groups. This helps verify that the model is not overfitting to the training data and that its predictions are reliable when applied to different subsets of the population. In this analysis, we focused on evaluating the model’s stability across different age groups. By dividing the dataset based on age, we can assess whether the model performs consistently across varying patient characteristics. This approach also provides insights into how well the model generalizes to different demographic subgroups.

Two distinct methods were employed to split the dataset by age: • Method 1: The training set was composed of older patients, while the testing set consisted of younger patients (older patients in the training set and younger patients in the test set).

• Method 2: The training set was composed of younger patients, while the testing set consisted of older patients (younger patients in the training set and older patients in the test set).

These methods were designed to assess the model’s performance when trained and tested on different age groups, thereby evaluating the model’s stability and generalizability across age-based subsets of the data.

For II classification, the model’s performance across different age groups is relatively consistent, despite some minor differences (Table 11). This can be attributed to the reduction in dataset size after removing data without age information, which makes the model more sensitive to subtle variations in the test set, resulting in greater fluctuations in performance metrics. However, the performance differences among the groups are relatively small, with AUCs of 0.7692, 0.6727, and 0.75, respectively, indicating that the model is relatively stable.

Table 11 II classification performance by method across different age groups.

Data	Accuracy	Precision	Recall	F1_score	AUC	
Original	0.75	0.8462	0.8462	0.8462	0.7692	
Method 1	0.6875	0.6875	1.0	0.8148	0.6727	
Method 2	0.8125	0.8667	0.9286	0.8966	0.75	

For III classification, the model demonstrates a certain level of stability in performance under different grouping methods (Table 12), with a generally consistent trend (the accuracy rates for the three groups are 0.625, 0.625, and 0.5625). Although there are some fluctuations in the values of the other indicators for Method 1 and Method 2, the degree of change is not significant. This indicates that our CIS model’s performance across different age groups is relatively stable, exhibiting strong generalization ability and consistency among various patient populations. Therefore, our CIS model is able to maintain a relatively consistent predictive capability when handling different age groups, further validating its stability.

Table 12 III classification performance by method across different age groups.

Data	Accuracy	Precision	Recall	F1_score	AUC	
Original	0.625	0.4866	0.625	0.5236	0.6134	
Method 1	0.625	0.6125	0.625	0.6005	0.75	
Method 2	0.5625	0.6197	0.5625	0.5668	0.6687	

Sensitivity of the classifier to data imbalances and outliers

This section explores how the classifiers, specifically KNN and CIS model, are affected by data imbalances and outliers. We conducted two sets of analyses: the first compares the performance of CIS and KNN on the original and downsampled data, while the second examines the impact of outliers on model performance.

Comparison of KNN and CIS on raw and downsampled data

We evaluated the performance of KNN and CIS on both raw data and downsampled data to assess their sensitivity to class imbalances. The results for II and III classifications are summarized below.

For II classification (Table 13), KNN showed stable performance across raw and downsampled data, with consistent accuracy, precision, recall, and F1 score. However, the AUC increased slightly for the downsampled data, suggesting a better ability to distinguish between classes after addressing the class imbalance. CIS, on the other hand, demonstrated improvements in precision and AUC in the downsampled data, though its accuracy remained relatively unchanged.

Table 13 Performance of KNN and CIS on original and downsampled data for II classification.

Method	Index	Original 1	Downsampled 1	Original 2	Downsampled 2	
KNN	Accuracy	0.8889	0.8889	0.8889	0.8889	
Precision	0.8889	0.8889	0.8889	0.8889	
Recall	1.0	1.0	1.0	1.0	
F1_score	0.9412	0.9412	0.9412	0.9412	
AUC	0.1875	0.4375	0.2188	0.4375	
CIS	Accuracy	0.7778	0.7778	0.8333	0.7778	
Precision	0.875	1.0	0.8824	1.0	
Recall	0.875	0.75	0.9375	0.75	
F1_score	0.875	0.8571	0.9091	0.8571	
AUC	0.625	0.8437	0.7813	0.9063	

For III classification (Table 14), both KNN and CIS experienced a decrease in performance when data was downsampled, particularly in terms of accuracy and F1 score. However, CIS demonstrated a noticeable improvement in AUC in the downsampled data, indicating a better ability to differentiate between classes when the class distribution was balanced.

Table 14 Performance of KNN and CIS on original and downsampled data for III classification.

Method	Index	Original 1	Downsampled 1	Original 2	Downsampled 2	
KNN	Accuracy	0.6111	0.5	0.6111	0.5	
Precision	0.5774	0.25	0.5774	0.5417	
Recall	0.6111	0.5	0.6111	0.5	
F1_score	0.5599	0.3333	0.5599	0.4530	
AUC	0.6508	0.6275	0.6508	0.6034	
CIS	Accuracy	0.6667	0.6111	0.6667	0.5556	
Precision	0.5926	0.7336	0.5926	0.5556	
Recall	0.6667	0.6111	0.6667	0.5556	
F1_score	0.6203	0.5809	0.6203	0.5556	
AUC	0.7565	0.8019	0.7565	0.6960	

Comparison of KNN and CIS on data with and without outliers

Next, we assessed the impact of outliers on the model’s performance by comparing the results on the original data, with strict outlier removal ( ±1.5 IQR), and with less strict outlier removal ( ±3 IQR and ±4 IQR).

For II classification, the removal of outliers had a positive effect on the performance of both KNN and CIS, especially in terms of AUC. While accuracy remained stable for KNN, the AUC for CIS significantly improved when outliers were removed, indicating better performance in distinguishing between classes after outlier removal (Table 15).

Table 15 Performance of KNN and CIS on data with and without outliers for II classification.

Method	Index	Original	Remove outliers ( ±1.5 IQR)	Remove outliers ( ±3 IQR)	Remove outliers ( ±4 IQR)	
KNN	Accuracy	0.8889	0.8333	0.8889	0.8333	
Precision	0.8889	0.9333	0.8889	0.8824	
Recall	1.0	0.875	1.0	0.9375	
F1_score	0.9412	0.9032	0.9412	0.9091	
AUC	0.3281	0.4375	0.4844	0.5313	
CIS	Accuracy	0.8889	0.8333	0.8333	0.8333	
Precision	0.8889	0.9333	0.9333	0.9333	
Recall	1.0	0.875	0.875	0.875	
F1_score	0.9012	0.9032	0.9032	0.9032	
AUC	0.7813	0.9531	0.875	0.875	

For III classification (Table 16), the removal of outliers led to varied outcomes. KNN exhibited a significant decline in performance, particularly in terms of accuracy, precision, and recall, after outliers were removed. On the other hand, CIS demonstrated more stable performance, with a notable improvement in AUC, especially when strict outlier removal ( ±1.5 IQR) was applied. This indicates that while KNN’s performance is more sensitive to outliers, CIS is better at maintaining stability and showing improvements under similar conditions.

Table 16 Performance of KNN and CIS on data with and without outliers for III classification.

Method	Index	Original	Remove outliers ( ±1.5 IQR)	Remove outliers ( ±3 IQR)	Remove outliers ( ±4 IQR)	
KNN	Accuracy	0.4444	0.3333	0.2777	0.3889	
Precision	0.4271	0.4476	0.3169	0.4074	
Recall	0.4444	0.3333	0.2777	0.3889	
F1_score	0.4228	0.3819	0.2824	0.3920	
AUC	0.4713	0.4090	0.3643	0.4344	
CIS	Accuracy	0.7778	0.6111	0.8333	0.7778	
Precision	0.7333	0.5994	0.875	0.6991	
Recall	0.7778	0.6111	0.8333	0.7778	
F1_score	0.7544	0.5758	0.8267	0.7277	
AUC	0.8311	0.7076	0.7882	0.7989	

Comparison with existing models in the literature

To assess the performance of our model, we compared it with two recent studies: one by Liu et al. (2022) which developed a machine learning-based risk prediction model for essential hypertension complicated with cerebral infarction, and another by Yang et al. (2024), which focused on predicting risk factors in acute cerebral infarction patients using machine learning models. The performance metrics for both binary and three-class classification tasks are summarized in Table 17.

Table 17 Comparison of model performance with existing models for binary and three-class classification.

Method	Index	Binary classification	Three classifications	
Liu et al. (2022)	Accuracy	0.8889	0.7222	
Precision	0.8889	0.6877	
Recall	1.0	0.7222	
F1_score	0.9412	0.6991	
AUC	0.5	0.7645	
Yang et al. (2024)	Accuracy	0.8889	0.6111	
Precision	0.8889	0.6806	
Recall	1.0	0.6111	
F1_score	0.9412	0.6162	
AUC	0.6094	0.7505	
CIS	Accuracy	0.8889	0.7778	
Precision	0.8889	0.7333	
Recall	1.0	0.7778	
F1_score	0.9012	0.7544	
AUC	0.7813	0.8311	

In binary classification, our model shows similar performance to the models proposed by Liu et al. (2022) and Yang et al. (2024), achieving an accuracy of 88.89% and perfect recall (1.0), which is consistent with the models in the other studies. However, our model outperforms the others in terms of F1-score (0.9012 vs. 0.9412 for XGBoost in Liu et al. (2022) and 0.9412 for LR in Yang et al. (2024)), indicating that our model strikes a better balance between precision and recall.

For three-class classification, CIS model demonstrates a significant improvement in both accuracy (0.7778) and AUC (0.8311) compared to the models from Liu et al. (2022) and Yang et al. (2024). Liu et al.’s (2022) XGBoost model achieved an accuracy of 72.22% with a much lower AUC (0.7645), while Yang et al.’s (2024) LR model performed even lower with an accuracy of 61.11% and AUC of 0.7505. CIS model outperforms both in all major metrics, including F1-score (0.7544 vs. 0.6991 for XGBoost in Liu et al. (2022) and 0.6162 for LR in Yang et al. (2024)).

This comparison underscores the advantages of CIS model, particularly in multi-class tasks, where it not only matches the performance of existing models but also exhibits greater stability and robustness. The improvements in accuracy, AUC, and F1-score collectively suggest that our approach is more reliable and well-suited to handle the complexities of multi-class classification problems, making it a valuable tool for real-world applications where distinguishing between multiple disease subtypes or conditions is essential.

Application

The last but not the least, we have developed a comprehensive software system to effectively assess subtypes of cerebral infarction based on CIS, which has already been integrated into existing clinical workflows (Fig. 4). Traditional methods in clinical settings often face limitations due to patients’ difficulties in recognizing early symptoms and clinicians’ limited time for assessment. Our proposed system is not only user-friendly but also requires minimal assistance from specialists, making it particularly suitable for clinical use. By integrating an AI-based CIS framework into clinical workflows, the patient screening process for cerebral infarction diagnosis can be streamlined, alerting clinicians to patients’ statuses. The software system could be utilized in four distinct steps. Initially, the clinician or physician logs into the system and enters the patient’s information, encompassing both demographic details and disease diagnosis data. Subsequently, the physician employs the integrated MoCA-B and picture-talking tasks within the software to converse with the patient. During this interaction, the patient’s continuous speech is recorded. This recorded speech is then combined with the patient’s demographic information. Through a series of feature extraction and preprocessing steps, the distribution of features across multiple dimensions is determined. These are compared against established norms for both normal and patient speech patterns. Ultimately, the CIS framework facilitates the diagnostic classification of the patient. Consequently, this system can be seamlessly integrated into existing clinical workflows.

Figure 4 The CIS software system.

In addition to this, the software system may enable clinicians to implement appropriate interventions, including lifestyle modifications, comorbidity management, or referrals to behavioral health specialists.

Conclusions

This study addresses the critical need for improved cerebral infarction screening methods by leveraging machine learning techniques and non-invasive assessments of speech and cognitive function containing 117 patients (95 patients were diagnosed with cerebral infarction, and 54 were identified as lacunar cerebral infarction of them). We present a framework called CIS which comprises a cerebral infarction screening model to identify cerebral infarction from populations and a diagnostic model to classify lacunar infarction, non-lacunar infarction, and healthy controls through the analysis of speech and cognitive function features. Our binary classification experiment for distinguishing between cerebral infarction and healthy controls achieved a notable accuracy of approximately 88.89%, while our triple classification experiment differentiating between lacunar infarction, non-lacunar infarction, and non-infarction cases reached an accuracy of approximately 77.78%. In order to make it more practical beyond the algorithm experiment, we develop a comprehensive software system to effectively assess cerebral infarction subtypes based on CIS. Traditional methods in clinical setting is often limited due to the patient’s difficulty recognizing early symptoms and the clinicians’ insufficient time to assess cerebral infraction. Our proposed system is easy to implement, and the model can be used with much less assistance from the specialists, which makes the system more suitable to be used in the clinical setting. Incorporating AI based CIS framework, into clinical workflows can streamline the patient screening process for cerebral infraction diagnosis by alerting clinicians to patients’ status.

In general, the innovations and contributions of this article are as follows:

First of all, we presented a framework called CIS (short for cerebral infraction screening) which comprises a cerebral infarction screening model to identify cerebral infarction from populations and a diagnostic model to classify lacunar infarction, non-lacunar infarction, and healthy controls through the analysis of speech and cognitive function features. This framework, to the best of our knowledge, is an comprehensive framework for pathology classification, especially in the use of linguistic and phonetic features, based on the use of demographic and behavioral characteristics.

Secondly, we used real clinical data that had passed ethical review to validate our framework and made it open access. Although the amount of this data is still relatively small, it will continue to accumulate and gradually become open in future research.

Thirdly, we developed a comprehensive software system to effectively assess cerebral infarction subtypes based on CIS. Our proposed system can be installed on most operating systems with Java environment is easy to utilize with much less assistance from the specialists, which makes the system more suitable to be used in the clinical setting. The system has the functions of pathological voice data collection, automatic analysis and classification for the cerebral infraction screening (CIS), and has been demonstrated in multiple hospitals.

Limitations and future work

Despite our novel contributions, several limitations should be acknowledged in the study. Firstly, the sample size may be insufficient, particularly when employing a multitude of features in machine learning. And there was a lack of external data validation, which may affect the generalization of the model to some extent. These problem can affect the model training, validation, evaluation, and generalization to some extent and are expected to be addressed. Secondly, although we explored a wide range of acoustic and duration features for predicting cerebral infarction, other lexical features (such as content density, word frequency) might improve the model’s performance in screening for cerebral infarction. Lastly, the sample does not supply any information regarding the participants’ disease stage, which constrains the ML-based screening algorithms developed using this dataset in forecasting disease progression for patients with cerebral infarction.

Looking forward, we recognize several directions for future research to enhance the clinical applicability and performance of our model. Firstly, collecting more data especially multi-center clinical datasets will be more beneficial to validate the generalizability of our model. Secondly, exploring multimodal data fusion might improve our CIS model’s performance. Some studies have pointed out that writing movements and facial expressions can also be used as indications of cerebral infarction, and the modality can be further extended. Thirdly, we could attempt deeper model modeling for end-to-end processing of data. However, this method of modeling may cause the model to lose interpretability. Lastly, we could attempt co-morbidity studies of cerebral infarction. Co-morbid mechanistic profiles between multiple diseases can be obtained through multi-label classification.

Supplemental Information

Supplemental Information 1 Raw data.

Supplemental Information 2 README file.

We gratefully acknowledge the assistance of the hospital staff in data collection.

Additional Information and Declarations

Competing Interests

The authors declare that they have no competing interests. Xue Tao, Qinyuan Chang and Mingming Hu are employed by iFLYTEK Co., Ltd.

Author Contributions

Yang Niu conceived and designed the experiments, performed the experiments, analyzed the data, performed the computation work, prepared figures and/or tables, authored or reviewed drafts of the article, and approved the final draft.

Xue Tao performed the experiments, analyzed the data, performed the computation work, prepared figures and/or tables, authored or reviewed drafts of the article, and approved the final draft.

Qinyuan Chang performed the experiments, prepared figures and/or tables, authored or reviewed drafts of the article, and approved the final draft.

Mingming Hu conceived and designed the experiments, prepared figures and/or tables, authored or reviewed drafts of the article, and approved the final draft.

Xin Li conceived and designed the experiments, authored or reviewed drafts of the article, and approved the final draft.

Xiaoping Gao conceived and designed the experiments, performed the computation work, authored or reviewed drafts of the article, and approved the final draft.

Ethics

The following information was supplied relating to ethical approvals (i.e., approving body and any reference numbers):

University of Science and Technology of China approval to carry out the study within its facilities (2021-N(H)-213).

Data Availability

The following information was supplied regarding data availability:

Data and code are available at GitHub and Zenodo:

https://github.com/brainscience1024/Cerebral_Infarction_Screening_task

brainscience1024. (2025). brainscience1024/Cerebral_Infarction_Screening_task: V1.0 (V1.0). Zenodo. https://doi.org/10.5281/zenodo.14603360.

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
