# Peer review of "Machine learning-based feature selection and classification for cerebral infarction screening: an experimental study"

_PeerJ Computer Science, doi:10.7717/peerj-cs.2704_

## Round 0.1 · original submission · Major Revisions

Based on the referee reports, I recommend a major manuscript revision. The author should improve the manuscript, taking carefully into account the reviewers' comments in the reports, and resubmit the paper.

Reviewer 1 ·

Basic reporting

The manuscript entitled “Machine learning-based feature selection and classification for cerebral infarction screening: an experimental study” has been investigated in detail. The manuscript presents an intriguing application of machine learning techniques for enhancing cerebral infarction screening through the analysis of speech and cognitive function assessments. The use of multiple feature selection methods and a variety of classifiers demonstrates a comprehensive approach to identifying the most effective models for classification tasks. The reported accuracies, particularly with the XGBoost classifier, indicate promising potential for clinical application. However, the study requires significant revisions to strengthen its contributions and ensure its findings are robust and generalizable. Detailed descriptions of the dataset, methodology, and experimental setup are necessary for reproducibility and validation of results. Additionally, comparative analyses with baseline models, statistical significance testing, and a thorough discussion of feature importance would enhance the manuscript's scientific rigor and clinical relevance. Addressing these areas will improve the manuscript's clarity, impact, and suitability for publication. Overall, with the suggested revisions, the manuscript has the potential to make a meaningful contribution to the field of medical diagnostics and machine learning applications in healthcare.
1) The manuscript introduces the application of machine learning techniques for cerebral infarction screening using speech and cognitive function assessments. However, it lacks a clear explanation of how this approach significantly differs from or improves upon existing methods in the literature. The authors should explicitly highlight the novel aspects of their methodology and its unique contributions to the field.
2) The paper mentions the use of a dataset comprising speech and cognitive function features from patients with lacunar and non-lacunar cerebral infarction, as well as healthy controls. However, it fails to provide detailed information about the dataset, such as the number of participants, the demographic distribution, the specific features collected, and how the data was preprocessed. Providing comprehensive details about the dataset is essential for assessing the study's validity and reproducibility.
3) While the manuscript lists the feature selection methods (LASSO regression, ReliefF, Random Forest) and classifiers (SVM, KNN, Decision Tree, Random Forest, Logistic Regression, XGBoost) used, it does not delve into the specifics of how these methods were implemented. The authors should describe the parameter settings, any cross-validation techniques employed, and how the models were trained and tested to ensure a thorough understanding of the process.
4) The study reports that XGBoost achieved the highest accuracy in both binary and ternary classification tasks. However, it does not compare these results against baseline models or other state-of-the-art approaches in cerebral infarction screening. Including comparisons with established methods would provide context for evaluating the effectiveness of the proposed approach.

Experimental design

5) The manuscript presents accuracy percentages for different classifiers but does not include statistical analyses to determine the significance of the results. The authors should perform and report statistical tests (e.g., paired t-tests, ANOVA) to validate whether the observed improvements are statistically significant.
6) Although the paper states that selected features significantly contributed to classification performance, it does not provide an in-depth analysis of which features were most important and why. Incorporating feature importance rankings and discussing their relevance to cerebral infarction subtypes would enhance the interpretability and clinical relevance of the findings.
7) The study does not address how well the proposed models generalize to unseen data or different populations. The authors should include experiments that test the models' robustness, such as using cross-validation, testing on external datasets, or evaluating performance across diverse patient groups to ensure the models are not overfitting to the training data.

Validity of the findings

8) The manuscript does not discuss the limitations of the current study or potential areas for future research. Addressing limitations, such as the small sample size or potential biases in the dataset, and suggesting directions for future studies would provide a balanced perspective and guide subsequent research efforts.
9) “Results and discussion” section should be editeded in a more highlighting, argumentative way. The author should analysis the reason why the tested results is achieved.
10) The authors should clearly emphasize the contribution of the study. Please note that the up-to-date of references will contribute to the up-to-date of your manuscript. The studies named- “A robust chaos-inspired artificial intelligence model for dealing with nonlinear dynamics in wind speed forecasting; Overcoming nonlinear dynamics in diabetic retinopathy classification: A robust AI-based model with chaotic swarm intelligence optimization and recurrent long short-term memory”- can be used to explain the methodology and feature selection in the study or to indicate the contribution in the “Introduction” section.
11) While the study suggests that incorporating machine learning techniques into clinical practice could improve early detection and diagnosis of cerebral infarction, it lacks a detailed discussion on how these models can be integrated into existing clinical workflows. The authors should elaborate on the practical steps required for implementation, potential challenges, and how their approach can be utilized by healthcare professionals.

·

Basic reporting

Manuscript ID 105147v1
This paper is related to reviewing the manuscript titled " Machine learning-based feature selection and classification for cerebral infarction screening: an experimental study"
The authors studied the use of machine learning approaches for feature selection and classification in speech and cognitive function evaluations to improve cerebral infarction screening.

Experimental design

Firstly, the proposed study is weak in terms of organization, presentation, content and results. Secondly, the productivity of the study, its contribution to science and its innovation are very weak. Standard and well-known machine learning algorithms have been applied to a dataset that contains very few data. The extent to which they have contributed to computer science is debatable.

Validity of the findings

1) In this study, the results are really insufficient, the findings and experimental analysis are very poor.

Additional comments

Therefore, my recommendation is reject.
Best regards.

Reviewer 3 ·

Basic reporting

The article is generally well-written and adheres to PeerJ standards in structure and language. The introduction effectively frames the motivation for the study, outlining the challenges in cerebral infarction screening and the potential benefits of using machine learning methods with non-invasive data. Relevant literature is cited, though the authors could improve the background by more explicitly highlighting the knowledge gaps this study addresses compared to existing work.
Minor improvements in language clarity would enhance readability, particularly in the methodology section, where technical terms and processes could be simplified for a broader audience. Additionally, a more cohesive transition between the introduction and methodology sections could further clarify the study’s objectives and approach.
Suggested Improvements:
- Expand the literature review to emphasize the novelty of using speech and cognitive data specifically for cerebral infarction screening.
- Refine language in certain sections to enhance clarity, particularly around feature selection and classifier descriptions.

Experimental design

The study is within the scope of PeerJ Computer Science, addressing an innovative application of machine learning in the medical screening of cerebral infarction. The methodology is largely sound and ethically conducted, with IRB approval and participant consent clearly documented.
The feature selection process (LASSO, ReliefF, Random Forest) and classifier choices are appropriate for this study. However, additional detail explaining the rationale for each feature selection method would strengthen transparency. Similarly, the authors could elaborate on preprocessing steps, especially regarding how missing data was handled, as this impacts reproducibility.
The evaluation metrics chosen (accuracy, precision, and F1 score) are appropriate but would be strengthened with the inclusion of recall and AUC-ROC to offer a more comprehensive view of classifier performance, especially for imbalanced data in the ternary classification task.
Suggested Improvements:
- Provide more detail on the rationale behind the chosen feature selection methods and classifier parameters.
- Describe the preprocessing steps in greater depth, focusing on missing data handling and data cleaning criteria.
- Add additional evaluation metrics, such as recall and AUC-ROC, for a fuller assessment of model performance.

Validity of the findings

The study presents valid findings that are consistent with the methods employed, showing promising results for binary and ternary classification tasks using machine learning. XGBoost’s performance highlights the strength of this classifier in handling complex features in the dataset. The results are well-supported, with clear tables and figures that detail feature importance and classifier performance.
The conclusions are appropriately limited to the results but could benefit from more discussion on limitations. For example, the authors mention the need for larger datasets to improve generalizability but should further discuss how the limited sample size might impact model robustness in varied clinical settings. Additional commentary on the sensitivity of certain models (e.g., KNN) to noise or imbalanced data would also strengthen the discussion.
Suggested Improvements:
- Expand on the limitations related to sample size and generalizability, discussing how they may impact clinical applicability.
- Provide more insight into classifier sensitivity to data imbalance and outliers, particularly for KNN.

Additional comments

The study makes a valuable contribution by exploring a non-invasive, machine-learning-based approach to cerebral infarction screening using speech and cognitive assessments. The authors have also shared their code and data publicly, which enhances the study's reproducibility and aligns with open science principles.

Reviewer 4 ·

Basic reporting

The grammatical aspect of the paper needs revision.
Other aspects of the paper are according to the guideline.

Experimental design

The paper has sufficient novelty and is recommended for publication.

Validity of the findings

The paper has sufficient novelty and is recommended for publication.

Additional comments

The paper has sufficient novelty and is recommended for publication. However, the grammatical aspect of the paper needs revision, and the authors need to revise the paper.

---

## Round 0.2 · accepted · Accept

The author has addressed the reviewer's comments properly. Thus I recommend publication of the manuscript.

Reviewer 1 ·

Basic reporting

All my comments have been thoroughly addressed. It is acceptable in the present form.

Experimental design

All my comments have been thoroughly addressed. It is acceptable in the present form.

Validity of the findings

All my comments have been thoroughly addressed. It is acceptable in the present form.

Reviewer 3 ·

Basic reporting

The manuscript is well-written, using clear and professional English. The introduction provides a detailed context for the study, highlighting the significance of cerebral infarction screening and the limitations of current methods. The literature is well-referenced, and the motivation for the study is clearly articulated. The structure adheres to PeerJ standards, ensuring clarity and coherence. I have no additional comments.

Experimental design

The experimental design is robust and aligns with the journal's scope. The authors have thoroughly described their methods, including feature selection, classification algorithms, and evaluation metrics. The inclusion of details about dataset preprocessing and cross-validation enhances reproducibility. The study meets high technical and ethical standards, with proper citations provided throughout. I have no concerns.

Validity of the findings

The authors present convincing evidence supporting their conclusions. The experiments are rigorous, with extensive comparisons against baseline models and ablation studies to validate the results. Statistical analyses strengthen the reliability of findings, and limitations and future directions are appropriately discussed. The conclusions are well-supported by the data and align with the stated objectives of the study.

Additional comments

The authors have addressed prior concerns effectively, and the revised manuscript reflects significant improvements. The inclusion of robustness testing across demographic subgroups and comparisons with state-of-the-art models enhances the study's validity and generalizability. I commend the authors for their thoroughness in responding to feedback and refining their manuscript.